# Expanding the olfactory code by in silico decoding of odor-receptor chemical space

**Sean Michael Boyle[1], Shane McInally[2], Anandasankar Ray[1,2,3]***

[1]Genetics, Genomics, and Bioinformatics Program, University of California, Riverside, Riverside, United States; [2]Department of Entomology, University of California, Riverside, Riverside, United States; [3]Institute of Integrative Genome Biology, University of California, Riverside, Riverside, United States

**Abstract** Coding of information in the peripheral olfactory system depends on two fundamental factors: interaction of individual odors with subsets of the odorant receptor repertoire and mode of signaling that an individual receptor-odor interaction elicits, activation or inhibition. We develop a cheminformatics pipeline that predicts receptor–odorant interactions from a large collection of chemical structures (>240,000) for receptors that have been tested to a smaller panel of odorants (~100). Using a computational approach, we first identify shared structural features from known ligands of individual receptors. We then use these features to screen in silico new candidate ligands from >240,000 potential volatiles for several Odorant receptors (Ors) in the *Drosophila* antenna. Functional experiments from 9 Ors support a high success rate (~71%) for the screen, resulting in identification of numerous new activators and inhibitors. Such computational prediction of receptor–odor interactions has the potential to enable systems level analysis of olfactory receptor repertoires in organisms.

*For correspondence: anand. ray@ucr.edu

## Introduction

The peripheral olfactory system is unparalleled in its ability to detect and discriminate amongst an extremely large number of volatile compounds in the environment. To detect this wide variety of volatiles, most organisms have evolved large families of receptor genes that typically encode 7-transmembrane proteins expressed in the olfactory neurons (*Buck and Axel, 1991*; *Clyne et al., 1999*; *de Bruyne and Baker, 2008*; *Vosshall et al., 1999*; *Dahanukar et al., 2005*). Each volatile chemical in the environment is thought to interact with a specific subset of odorant receptors depending upon odor structure and binding sites on the receptor. This precise detection and coding of odors by the peripheral olfactory neurons are subsequently processed, transformed and integrated in the central nervous system to generate specific behavioral responses that are critical for survival such as finding food, finding mates, avoiding predators etc (*van der Goes van Naters and Carlson, 2006*).

Currently there are two major rate-limiting steps in analysis of peripheral coding in olfaction: a very small proportion of chemical space can be systematically tested for its activity on odorant receptors and a very small fraction of the numerous odorant receptors have been tested for responses (*Araneda et al., 2000*; *Hallem et al., 2004*; *Hallem and Carlson, 2006*; *Pelz et al., 2006*; *Kreher et al., 2008*; *Saito et al., 2009*; *Mathew et al., 2013*). The challenges for overcoming the rate-limiting steps are enormous. First, volatile chemical space is immense, more than 2000 odors in the environment have been catalogued from a small fraction of plant sources alone (*Knudsen et al., 2006*). Second, the complete three-dimensional structures of the 7-transmembrane odorant receptor proteins have not yet been determined and modeling of protein–odor interactions and sophisticated virtual screening methods are not yet possible except in rare instances (*Triballeau et al., 2008*). In the decade since the first systematic study of 47 odorants on the *Drosophila* antenna in 2001 (*de Bruyne et al., 2001*), additional studies have only identified a total of ~250 novel activating odors (*de Bruyne et al.,*

**eLife digest** Although our sense of smell is regarded as inferior to that of many other species, we can nevertheless distinguish between roughly 10,000 different odors. These are made up of molecules called odorants, each of which activates a specific subset of odorant receptors in the nose. However, much of what we know about this process has come from studying the fruit fly, *Drosophila*, which detects odors using receptors located mainly on its antennae.

The number of potential odorants in nature is vast, and only a tiny fraction of the interactions between odorants and receptors can be physically tested. To address this challenge, Boyle et al. have used a computational approach to study in depth the interactions between a subset of 24 odorant receptors in *Drosophila* antennae and 109 odorants.

After developing a method to identify structural features shared by the odorants that activate each receptor, Boyle et al. used this information to perform a computational (in silico) screen of more than 240,000 different odorant-like volatile compounds. For each receptor, they compiled a list of the 500 odorants predicted to interact most strongly with it. They then tested their predictions for a subset of the receptors by performing experiments in living flies, and found that roughly 71% of predicted compounds did indeed activate or inhibit their receptors, compared to only 10% of a control sample.

In addition to providing new insights into the nature of the interactions between odorants and their receptors, the computational screen devised by Boyle et al. could aid the development of novel insect repellents, or compounds that mask the odors used by disease-causing insects to identify their hosts. It could also be used in the future to develop novel flavors and fragrances.

*1999*; *de Bruyne et al., 2001*; *Dobritsa et al., 2003*; *Goldman et al., 2005*; *Hallem et al., 2004*; *Hallem and Carlson, 2006*; *Kreher et al., 2005, 2008*; *Kwon et al., 2007*; *Pelz et al., 2006*; *Stensmyr et al., 2003*; *Turner and Ray, 2009*; *van Naters and Carlson, 2007*; *Yao et al., 2005*; *Schmuker et al., 2007*), which have been assembled and compared in an online database (*Galizia et al., 2010*).

Here we overcome this challenge by designing a chemical-informatics platform that is effective and fast. In order to do so we focused our attention on one of the most comprehensive quantitative data sets available, where measurements of responses of 24 *Drosophila* odorant receptors to a panel of 109 odorants are known that provides a rich resource for structure-activity type analyses (*Hallem and Carlson, 2006*). We devised a method to identify molecular structural properties that are shared amongst the activating odorants for each receptor. We then utilize information about these shared molecular features of active odorants, that are presumably required for binding to a receptor, to perform in silico screens on a chemical space of >240,000 chemicals, including a large collection of naturally occurring and biologically important odors, and identify the top 500 hits for each of the odorant receptors (Ors). We then use single-unit electrophysiology to validate a subset of predictions for 9 Ors in vivo and find that our method met an overall success rate of ~71% in identifying novel ligands. This approach is specific since testing shows a low (10%) rate of finding ligands while using non-predicted odors. This approach allows us to create a computationally predicted peripheral coding map of a large chemical space, which substantially improves our ability to predict and investigate peripheral olfactory coding and provides a powerful tool for the discovery of novel ligands for Ors, some of which may be ecologically important or useful for behavior modification.

## Results

### Analysis of odorant structure

Since the structure of receptor protein complexes is not known, we analyzed receptor–odor interactions by applying the 'similarity property principle', which reasons that structurally similar molecules (e.g., activating odorants) are more likely to have similar properties (*Hendrickson, 1991*; *Martin et al., 2002*). Although this general approach has been useful in the area of pharmaceuticals (*Martin et al., 2002*; *Keiser et al., 2009*), receptor–odor analysis presents significant additional challenges. Not only are odorant molecules generally smaller in size than pharmaceuticals (average MW of known odors ~threefold less than FDA approved pharmaceuticals [*Wishart et al., 2008*]) and therefore offer fewer

structural features for differentiation, they are also detected by the receptors with specificity at extremely low concentrations in the volatile phase (*Hallem and Carlson, 2006*; *Kreher et al., 2008*). Additionally, odorant receptors are differentially tuned and can sometimes appear not to follow distinct structural rules: odors that look structurally different can strongly activate the same receptor, while odors that appear very similar may have very different levels of activity (*Hallem and Carlson, 2006*). For example, while hexanal and γ-octalactone are structurally very different, they both strongly activate Or85b (*Hallem and Carlson, 2006*). Alternatively, while hexanal and pentanal are structurally very similar, they have very different activities against Or85b (*Hallem and Carlson, 2006*).

## General measures of odorant similarity

Similarity in chemical structure can be described and measured quantitatively using multiple approaches, however a single method may not be ideal for every single application (*Maldonado et al., 2006*). In order to test whether non-optimized approaches would be able to identify similarities in shape of known activators we compared four different approaches: Cerius2 (Accelrys Software Inc), Dragon (Talete), Maximum-Common-Substructure (MCS) (*Cao et al., 2008b*), and atom-pair (AP) (*Carhart et al., 1985*; *Cao et al., 2008a*). Cerius2 and Dragon represent collections of 200 and 3224 molecular descriptors, respectively, that calculates values for a broad range of chemical properties such as molecular weight, functional group counts, and in the case of Dragon, three-dimensional relationships within molecules. The AP method compares shortest path distances between all atom pairs in a molecule. Lastly, MCS identifies the largest two-dimensional substructure that exists between two compounds. Using each of these approaches, we computed distances between 109 odors that had previously been tested against 24 Ors from *Drosophila melanogaster* (*Hallem and Carlson, 2006*). These represent most of the *Or* genes expressed in the *Drosophila* antenna (*Hallem and Carlson, 2006*). Upon comparison, we find that none of the four methods were vastly superior and that each method varied in the ability to group known activating odorants 'actives' close together in distance as measured for each Or using a method called accumulative-percentage-of-actives (APoA)(*Chen and Reynolds, 2002*) ('Materials and methods' and *Figure 1—figure supplement 1*) and value of the area-under-the-curve (AUC). Ultimately, Dragon and Cerius2, which utilize a large number of diverse molecular descriptor values to describe each odor structure, performed better than AP or MCS, suggesting that a more diverse set of descriptors is better at explaining Or activity than two-dimensional measures alone (*Figure 1B*). Atom-Pair and MCS were subsequently ignored from further development.

## Identification of unique subsets of optimized descriptors for each *Drosophila* Or

Individual Ors respond to distinct subsets of ligands with some degree of overlap (*Hallem and Carlson, 2006*; *Kreher et al., 2008*). We reasoned that rather than using entire Dragon or Cerius2 descriptor sets, which likely includes a number of measurements for features irrelevant for ligand-binding to an individual Or, judiciously selecting subsets of molecular descriptors suited to cluster activators for an individual receptor may be more effective at defining an Or-specific chemical space. To test this hypothesis, we used a Sequential-Forward-Selection (SFS) method to incrementally create unique optimized descriptor subsets for each Or from an initial combined set of 3424 descriptors from Dragon and Cerius2 (*Whitney, 1971*) ('Materials and methods'; *Figure 1A*). This optimization-based analysis was performed on the 19 Ors from the dataset with known activating odors, excluding Or82a, since it has but a single known strong activator (*Hallem and Carlson, 2006*).

Not surprisingly, the composition of the optimized descriptor sets varied greatly between Ors, as on average only 13% of descriptors are shared between Ors (*Table 1*; *Supplementary file 1A*). Molecular descriptors can be categorized from 0 to 3 dimensions. Zero-dimensional (0-D) descriptors define features that can be viewed as not directly being shape dependent, such as molecular weight or vapor pressure. On the other end of the scale, three-dimensional (3-D) descriptors define features of molecules in three-dimensional space, such as the distance between two atoms of an odor molecule. Interestingly, we find an overwhelming preference for three-dimensional and two-dimensional descriptors compared to one-dimensional and zero-dimensional descriptors, suggesting that structural shape features are more important for receptor–odor interactions (*Table 1*; *Supplementary file 1A*). We find that Or-optimized descriptor sets were far superior at grouping together activating odors from the training set when compared to the non-optimized methods (Dragon, Cerius2, MCS, AP) and a previously identified collection of descriptors that were identified without receptor-specific optimization (*Haddad et al., 2008*) (*Figure 1B*, *Figure 1—figure supplement 1*).

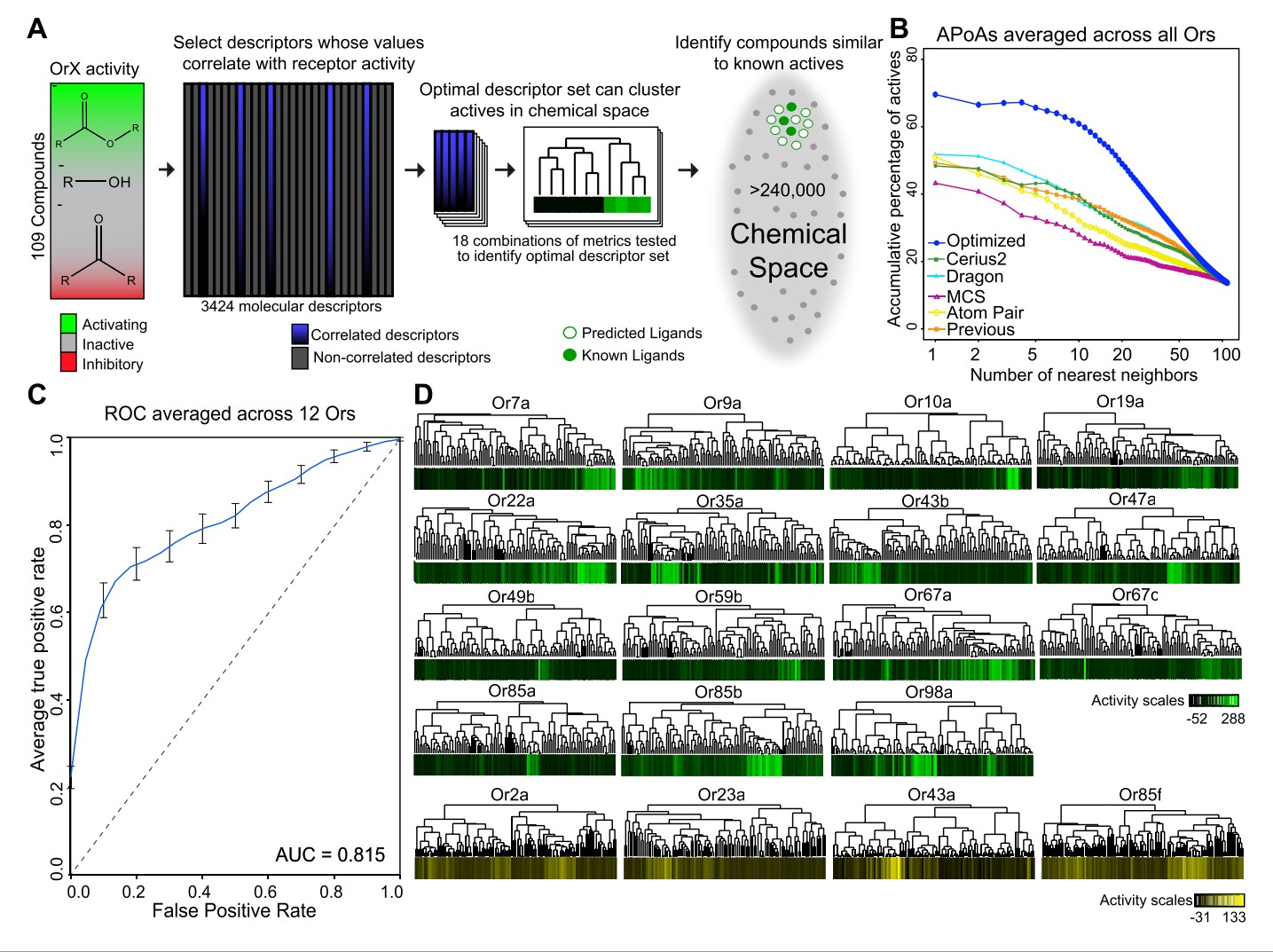

**Figure 1**. A receptor-optimized molecular descriptor approach has strong predictive power to find new ligands. (**A**) Schematic of the cheminfomatics pipeline used to identify novel ligands from a larger chemical space. (**B**) Plot of mean APoA values for 19 Drosophila Ors calculated using various methods including a previously identified set (*Haddad et al., 2008*). (**C**) Receiver-operating-characteristic curve (ROC) representing computational validation of ligand predictive ability of the Or-optimization approach. (**D**) Hierarchical cluster analysis of the 109 odorants of the training set using Or-specific optimized descriptor sets to calculate distances in chemical space for odorant receptors with strong activators (green), and odorant receptors with no strong activators (yellow).

The following figure supplements are available for figure 1:

**Figure supplement 1**. Analysis of APoA curves for individual odor receptors.

**Figure supplement 2**. Pharmacophores of active compounds for individual Ors.

## Computational validation of optimized descriptor sets

In order to validate the predictive ability of the *Or*-optimized method, we performed five independent trials of fivefold cross-validations followed by a Receiver-Operating-Characteristic (ROC) analysis, an established computational approach (*Hastie et al., 2001*; *Tan et al., 2006*) ('Materials and methods'). Briefly, this involved withholding 20% of the 109 previously tested odors for a receptor. Descriptors were optimized using the remaining 80% odors for training, and ligand-predictions were subsequently performed on the 20% of odors that were withheld. This operation was repeated five times for each receptor, each time selecting a different 20% as withheld from the training set. The entire fivefold

**Table 1.** Optimized molecular descriptor set compositions

| Descriptor class type counts for all Ors | |
| --- | --- |
| GETAWAY descriptors | 75 |
| 3D-MoRSE descriptors | 66 |
| 2D autocorrelations | 44 |
| Edge adjacency indices | 44 |
| 2D binary fingerprints | 44 |
| Functional group counts | 43 |
| Atom-centred fragments | 37 |
| WHIM descriptors | 36 |
| Topological charge indices | 24 |
| Atomtypes (Cerius2) | 23 |
| Burden eigenvalues | 23 |
| Molecular properties | 23 |
| Topological descriptors | 22 |
| Geometrical descriptors | 18 |
| 2D frequency fingerprints | 11 |
| RDF descriptors | 8 |
| Walk and path counts | 6 |
| Connectivity indices | 5 |
| Information indices | 5 |
| Topological (Cerius2) | 4 |
| Constitutional descriptors | 3 |
| Structural (Cerius2) | 2 |
| Randic molecular profiles | 2 |
| **Optimized descriptor analysis** | |
| Average descriptor overlap between Ors | 13% |
| Average number of descriptors per Or | 29.9 |
| Average number 3D descriptors per Or | 10.8 |
| Average number 2D descriptors per Or | 12.2 |
| Average number 1D descriptors per Or | 6.6 |
| Average number 0D descriptors per Or | 0.3 |
| **Descriptor dimensionality counts** | |
| Number three dimensional descriptors | 205 |
| Number two dimensional descriptors | 232 |
| Number one dimensional descriptors | 126 |
| Number zero dimensional descriptors | 5 |
| **Descriptor Origin** | |
| Number Dragon descriptors | 539 |
| Number Cerius descriptors | 29 |

Breakdowns of the molecular descriptor class type, dimensionality, origin, and average overlap for all optimized molecular descriptors selected for each Or.

operation was repeated five times for each receptor and a mean ROC curve representing the prediction accuracy determined. This analysis was possible for 12 *Ors* which had >6 known ligands that activated >100 spikes/s. The Area-Under-Curve (AUC) value (0.815) is very promising and suggests that the *Or*-optimized descriptor sets are effective at predicting novel ligands (*Figure 1C*).

In addition to performing the fivefold cross-validation, we also clustered the 109 training odors independently for each Or, using distances calculated from the previously determined receptor specific descriptor sets we identified. As expected, we find that activating odorants cluster tightly together for each Or (*Figure 1D*) and activating odors of an Or have shared sub-structures and shared pharmacophore features (*Figure 1—figure supplement 2*). In a few cases, such as for Or35a and Or98a, not all the highly activating compounds are clustered, suggesting the possibility of multiple or flexible binding sites, or imperfect selection of descriptors. Four of the Ors (Or2a, Or23a, Or43a and Or85f) have few known activators, none of which activate the receptors at >150 spikes/s, however our descriptor optimization approach is still able to cluster each of the few weak activators together (*Figure 1D*).

## High-throughput in silico screening of odorant receptors

Since Or-optimized descriptor sets can efficiently group strong activators in chemical space, we used them to rank untested compounds according to their distance from known activators for specific Ors. We assembled a natural odor library, which contains 3197 naturally occurring odors, and a library derived from Pubchem (*Bolton et al., 2008*), which contains >240,000 compounds with similar molecular weights and atom type compositions to known volatiles ('Materials and methods'). We then systematically screened both libraries using the optimized descriptor sets of 19 *D. melanogaster* Ors in silico. We identify the top 500 (0.2%) hits from this vast chemical library for each Or, the top ~100 of which are reported in *Supplementary file 1B*.

## Electrophysiological validation of in silico screen and identification of agonists

To validate our in silico screen, we obtained a large number of untested odorants belonging to the top 500 predicted ligands for nine different Ors (141 total interactions tested; ~11–23/Or) that were available from commercial sources at high purity and reasonable prices. The nine receptors were selected on the basis of previous functional mapping studies that enable us to unambiguously identify the antennal olfactory

receptor neurons (ORNs) they are housed in (*Hallem et al., 2004*; *Couto et al., 2005*). We systematically tested each predicted receptor–odor combination using single-unit electrophysiology to record from the ORNs to which these 9 Ors have been previously mapped (*Hallem et al., 2004*; *Couto et al., 2005*). We find that a majority of the predicted ligands evoked responses from the target ORNs; ~71% evoked either activation (>50 spikes/s above the spontaneous activity) or inhibition (>50% reduction in spontaneous activity [reverse agonist activity]) (*Table 2*). These cutoffs were selected based on the study from which the training set was obtained and has been used in other studies in the past that use this type of recordings (*Hallem and Carlson, 2006*; *Kreher et al., 2008*). Interestingly, the mean vapor pressure of activating odors (11.84 Torr) is 7.5 times higher than of inactive odors (1.58 Torr), raising the possibility that some inactive odors may not be volatilized and delivered at adequate levels to the ORNs. Additionally, we find that ~13% of the predicted compounds we tested showed an inhibitory effect on baseline activity of the respective neuron (*Table 2*). These inhibitors were identified by virtue of structural similarity to known activators suggesting that they may bind to similar sites on the receptor. Thus as an additional benefit our approach may provide a method to identify inhibitors as well. Such inhibitors would not only provide important tools to investigate mechanisms of odorant receptor inhibition but could also be used in blocking specific odor-mediated behaviors. Consistent with our observations three of the receptor–odor interactions had been previously identified independently as well, Or22a (*Pelz et al., 2006*), and Or49b (*Hallem et al., 2004*). The electrophysiological analysis provides the most important validation of our Or-optimized descriptor-based in silico screen.

## Odor response spectra of individual Ors

Since we systematically analyzed responses of a large number of new odorants individually, we were able to characterize the odor-response spectra of several antennal ORN classes to these new ligands (*Figure 2A*). New activators are reported for every receptor, and inhibitors are identified for several. Ligand predictions for 2 of the 3 receptors that do not perform as well are Or10a and Or49b that detect aromatic compounds. Their poor performance is explained by the lack of aromatic ligands in the initial training set (13/109) odorants. We find that a >85% of the predicted ligands activate odorant receptors Or7a, Or22a, Or59b, Or85a, Or85b, and Or98a (*Figure 2A*).

## Specificity of in silico predicted ligands

We rigorously examined the rate of false negative predictions for each Or by systematically testing newly identified ligands of each Or against the other non-target receptors using electrophysiology. Of 504 non-target receptor–odor interactions tested, we found that only 10% evoked a response >50 spikes/s and 3.7% evoked a response >100 spikes/s (*Figure 3A*). This represents a high degree of specificity, especially considering that the Or-optimized descriptor method did not incorporate any additional computational screening to rule out non-target activators. Additionally, when we plot the percentage of odors that validated as activators when tested using electrophysiology (considering both predicted and non-target receptor–odor interactions), we find that activity is strongly related to predicted odor ranking (*Figure 3B*). Odors which rank closest to known activators for each Or, particularly within the top 500 hits, are far more likely to be activators than odors further away, and there is a drastic drop-off in activating odors present beyond the 1000 rank. We see the same trend if we plot mean activity of odors for the same ranking divisions. Highly ranked odors have a far higher mean activity than distantly ranked odors.

**Table 2.** Predicted receptor–odor interactions validated as highly accurate using electrophysiology

| Classification | Or7a | Or10a | Or22a | Or47a | Or49b | Or59b | Or85a | Or85b | Or98a | Total |
|---|---|---|---|---|---|---|---|---|---|---|
| Ligands (%) | 88 | 31 | 86 | 39 | 27 | 91 | 92 | 87 | 100 | 71 |
| Agonists (>50 spikes/s) (%) | 63 | 31 | 81 | 33 | 18 | 64 | 69 | 70 | 92 | 58 |
| Agonists (>100 spikes/s) (%) | 31 | 13 | 62 | 11 | 9 | 45 | 48 | 48 | 67 | 37 |
| Inverse agonists (%) | 25 | 0 | 5 | 6 | 9 | 25 | 23 | 17 | 8 | 13 |

Summary of prediction accuracy percentages obtained by electrophysiology validation. Ligands = Agonists (≥50 spikes/s) + Inverse agonists (>50% reduction from baseline activity).

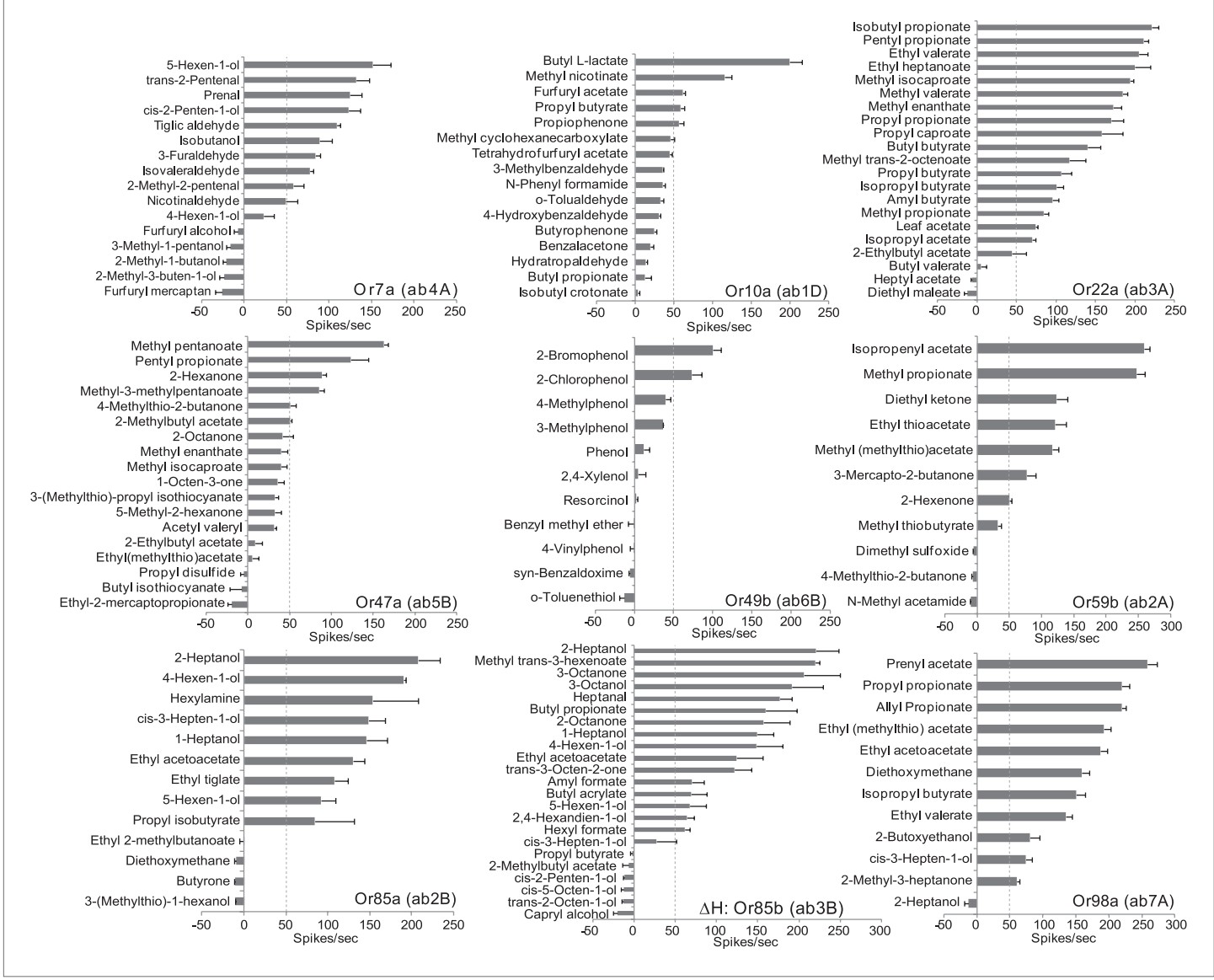

**Figure 2**. Electrophysiology validates that odorant receptor-optimized molecular descriptors can successfully identify new ligands for Drosophila. Mean increase in response of neurons to 0.5-s stimulus of indicated odors (10$^{-2}$ dilution) predicted for each associated Or. Dashed lines indicate the activator threshold (50 spikes/s). ΔH: Or85b (ab3B) = flies lack expression of Or22a in neighboring neuron, thus all observed neuron activation is unambiguously caused by Or85b. N = 3, error bars = s.e.m.

## Relationship between descriptor sets and Or sequence and activity

Since receptor-optimized descriptor sets and the predicted ligand space they define are a function of shared molecular features that a receptor may employ to recognize ligands, we were now in a position to determine how these characteristics correlate with receptor properties such as their known-activity profiles and amino acid sequences. We used hierarchical cluster analysis to create trees that represent the various receptors based on: shared descriptors selected; known activity-based relationships (*Hallem and Carlson, 2006*); degree of overlap of predicted ligands; and amino acid sequence (*Figure 4A*; 'Materials and methods'). We found that the maximum overlap in Or relationships is retained between the descriptor and the activity trees, and the descriptor and the cross activity trees with 11 out of 24 Ors present in subgroups that are common in both cases. However, only two subgroups (yellow and grey) are conserved across the three trees. The largest shared overlap existing in the descriptor tree suggests that the Or-optimized descriptors

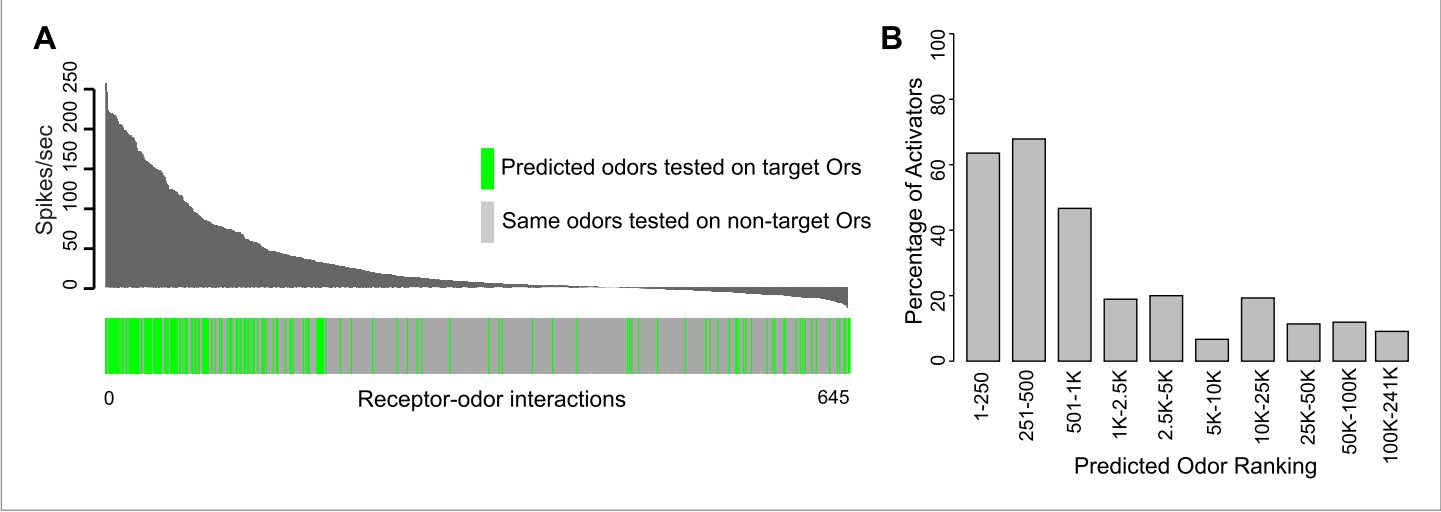

**Figure 3**. Predicted receptor–odor interactions are highly specific. (**A**) Plot of activity (Top) for electrophysiologically tested receptor-odor interactions. (Bottom) Plot indicating locations of predicted receptor-odor combinations (green) and same odorants tested in non-target receptor-odor combinations (gray). (**B**) Plot of percentage of activating odors (>50 spikes/s) considering all activating or inactive odors (>0 spikes/s) across ranking bins for all odors tested using electrophysiology.

link the known and the predicted receptor–odor interactions and that our analysis may expand upon odorant receptor activity relationships beyond those previously known from the training data. We also found that the phylogenetic tree has fewer relationships conserved with each of the trees, consistent with previous observations (*Hallem et al., 2004*) supporting the idea that, while the most conserved amino acid residues in the Ors provide the structure of the tree, they do not correlate strongly with ligand specificity.

### Analysis of breadth of predictions for each Or in chemical space

Coding of odors in a large volatile space (>240,000) by a receptor repertoire is virtually impossible to determine experimentally. However, based on the Or-optimized descriptor sets we computationally derived prediction frequency distributions for each of the *Drosophila* Ors in this large chemical space (*Figure 4B*). As expected, we find substantial variation in the distribution frequency of predicted ligands across different receptors. The predicted response profiles support previous observations made with smaller odor panels that the olfactory system can potentially detect thousands of volatile chemicals, many of which the organism may never have encountered in its chemical environment. Plant volatiles constituted a large portion of compounds that are predicted to be ligands for *Drosophila* Ors. To further analyze odor source representation, we classified odors that belong to top 500 prediction lists according to their source, if known, and find that Ors are not specialized for odors from a single source (*Figure 5A*).

### Across-receptor activation patterns in *Drosophila*

To study the ensemble activation patterns of odors predicted across all Ors, we analyzed the across-receptor activation patterns of the 3197 known compounds for nine receptors (Or7a, 10a, 22a, 47a, 49b, 59b, 85a, 85b, 98a). Surprisingly, we find that only 25% of compounds from the 'natural' odor library found in the top 500 predictions for each Or are predicted to activate multiple Ors (*Figure 5B*, lower left panel). If we consider compounds from the Pubchem library in the top 500 predicted activators for each receptor, we observe further reduction in the proportion of across-receptor activating compounds (*Figure 5B*, upper right). Consistent with this prediction we find that cross-activation by ligands functionally evaluated in this study for nine receptors is lower than that reported previously using ligands of comparable strength for the same nine receptors (*Hallem and Carlson, 2006*) (*Figure 5B*, lower right panel). These data suggest that a number of natural odors may be detected by a few receptors, particularly at low concentrations.

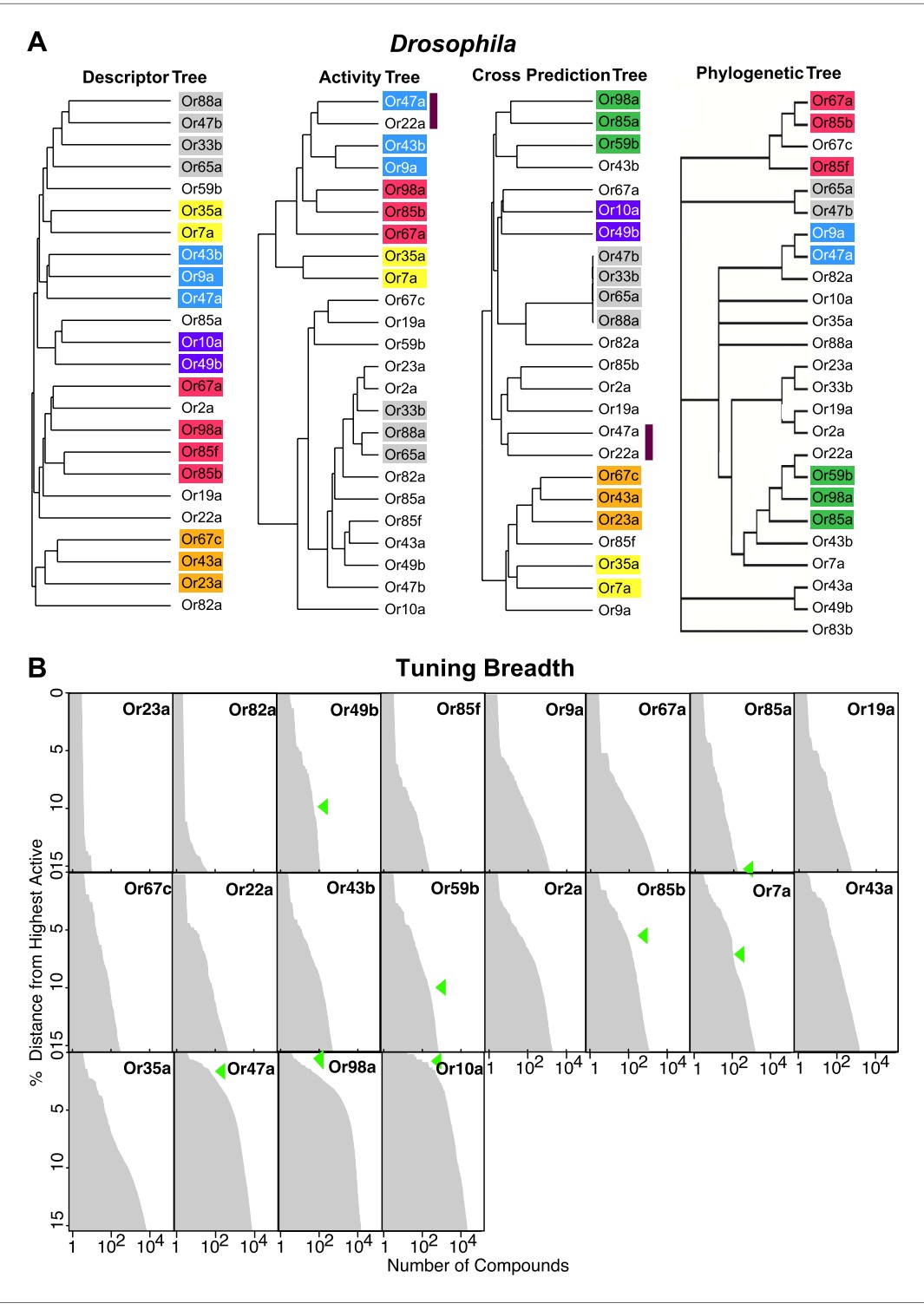

**Figure 4**. Analysis of receptor–odor relationships and breadth of tuning. (**A**) Hierarchical clusters created from Euclidean distance values between Drosophila Ors calculated using: (left to right) shared optimized descriptors; known activity to training set odors (**Hallem and Carlson, 2006**); overlap across top 500 predicted ligands; and Phylogenic tree of receptors (**Hallem and Carlson, 2006**). Sub clusters shaded with colors or bars. (**B**) Frequency distribution of compounds from the >240K library within the top 15% distance from highest active plotted to generate predicted breadth of tuning curves. Green arrows indicate relative distance of the furthest known activating compound determined by electrophysiology.

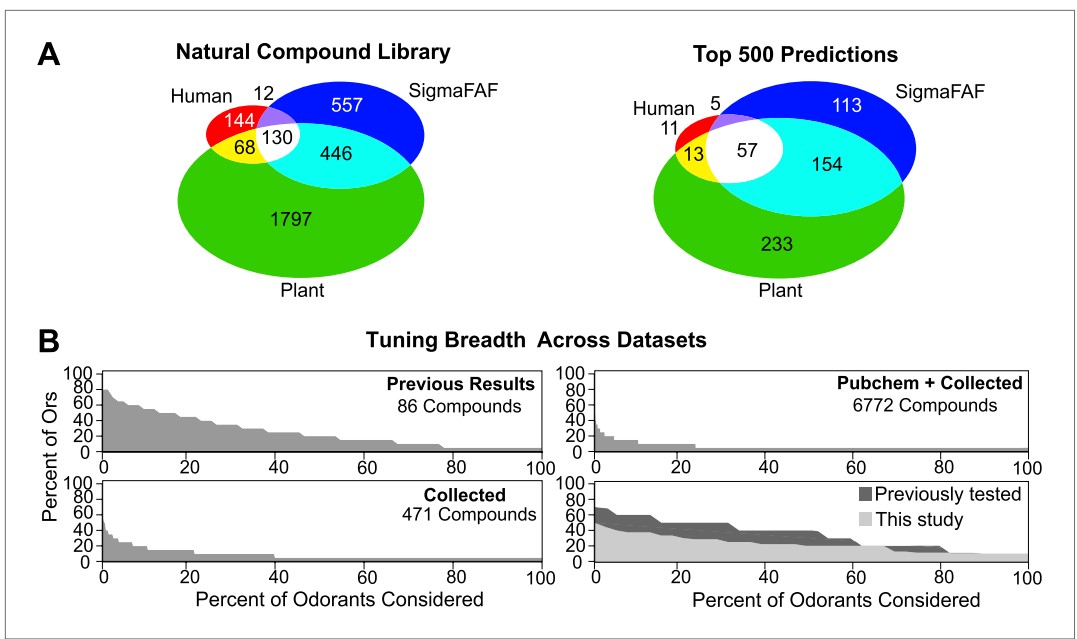

**Figure 5**. Analysis of predicted natural odor sources and cross activation. (**A**) (Left) The numbers of compounds present in the collected volatile library according to source. (Right) The numbers and sources of predicted ligands for the 19 Drosophila odor receptors/neurons within the top 500 predicted compounds. (**B**) Comparison of plots for percentage of receptors that are: (top left) activated by percentage of known odors from training set (***Hallem and Carlson, 2006***); (bottom left) predicted to be activated by Natural compound library; (top right) predicted to be activated from >240K library; and (bottom right) activated by ligands for 10 shared *Ors* in this study vs activated by comparable actives previously tested (***Hallem and Carlson, 2006***).

## Discussion

A primary element of the olfactory code is information about odor identity, represented by the characteristic interaction of an odor with the ensemble of olfactory receptors in the nose. Here we report an in silico approach to systematically identify ligands from a vast chemical space for a large number of Ors expressed in the antenna of *Drosophila*. We demonstrate that our predictions are accurate using two different validation approaches—computational validations and functional validation using electrophysiology. There is a strong correlation between ranks of predicted ligands to electrophysiological activity.

Obtaining and testing odors using traditional methods is time and cost intensive. Electrophysiology and calcium imaging are consuming processes that require not only a great deal of time to perform, but also the purchase of each odor to be physically tested. Moreover, large plate-based combinatorial chemical libraries, which are commonly implemented in drug discovery in the pharmaceutical industry, are not available for volatile odor libraries at reasonable costs. Since *Drosophila* is a premier model for understanding neurobiology of olfaction, several laboratories over the last 12 years have together screened ~250 odors, activities of which have been and compiled into a valuable database that standardizes across studies (***Galizia et al., 2010***). In this study we screen >240,000 chemicals and predict >10,000 new ligands which represents a substantial expansion of the known peripheral olfactory code for this important model organism and provides a system-level view of odor detection (***Figure 6A***).

The predicted ligands and prediction method will increase the speed of receptor–odor decoding and allow for interpretation of data at a large scale that is difficult to achieve. This could help answer questions such as breadth of receptor tuning, investigating responses to odorants from natural sources, and evolution of odor coding across a receptor repertoire. Additionally, using chemical informatics, it becomes possible to infer and prioritize for testing the network of odorant receptors that are activated from complex odor blends without the expensive and time consuming process of purchasing and testing all possible odors and receptor combinations (***Figure 6B***).

Interestingly, our attempts to identify molecular descriptors that would differentiate agonists from inverse agonists were not successful with this data set. This could be due to several reasons:

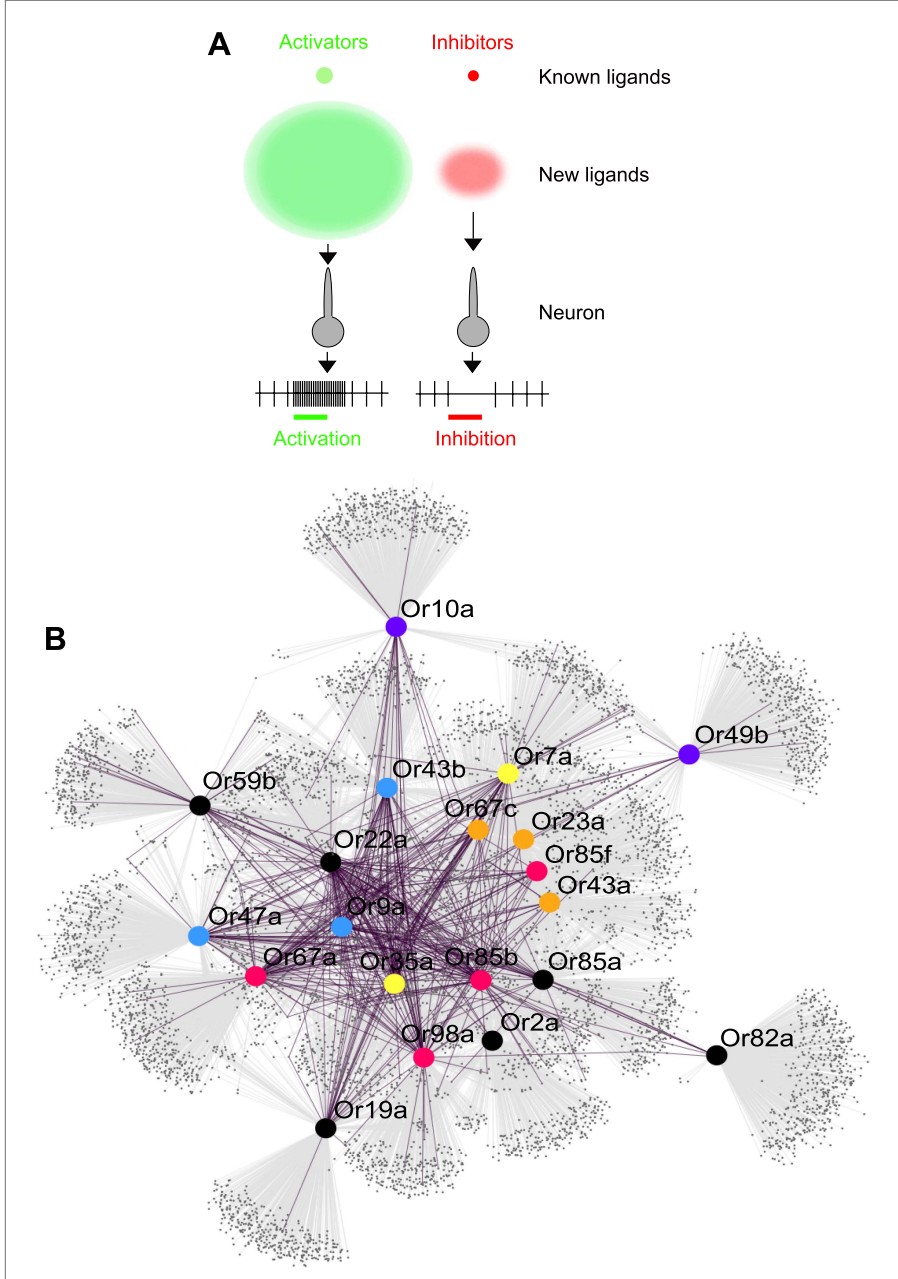

**Figure 6**. Predicted odor space and network view of odor coding. (**A**) Expansion of the peripheral olfactory code in this study: large increase in numbers of identified activators and inhibitors. The different sized circles represent the approximate ratio of numbers of previously known ligands (top circles), predicted ligands based on a cutoff of the top 500 predicted compounds per receptor and corrected to the validation success rate (lower, diffuse circles). (**B**) *Drosophila* receptor–odor network. Each known interaction (>50 spikes/s) from this and previous studies (*Hallem and Carlson, 2006*) is linked by a purple edge. Predicted receptor–odor network (top 500 hits) are linked by light-grey edges. All compounds are represented as small black circles and Ors are represented as large colored circles matching the colors used in (*Figure 4A*).

an insufficient number of inverse agonists amongst the training odors, or the inverse agonists may act via the same binding sites as agonists and share many of the same structural features of the activating odors making them difficult to distinguish. We feel that this remains an important challenge to be overcome in the future with improved computational approaches or larger odor training sets.

A similar, yet much smaller, analysis applied chemical informatics on *Drosophila* olfactory neuron activities to 47 odorants and screened ligands from 21 untested compounds in *Drosophila* (*Schmuker*

*et al., 2007*). Although this study had a relatively modest success rate of ~25% at predicting untested odorants as activators (by applying the same 50 spikes/s threshold for comparison), it also highlighted that structure-based ligand prediction is a viable method for further development. In another interesting analysis a Quantitative Structure Activity Relationship (QSAR) model was applied to describe odor-activity for *Drosophila* Ors. Using cheminformatics, important amino acid residues were identified using information from orthologous Or sequences identifying potential odor-binding regions, which was postulated to be 15 angstroms deep and 6 angstroms wide (*Guo and Kim, 2010*). These studies, along with ours, suggests that computational approaches could have great utility in study of sensory receptors. It will also be very interesting to use our method for making ligand predictions for the structurally distinct receptors such as olfactory ionotropic glutamate receptors (IRs), and gustatory receptors (Grs) in insects, and olfactory and taste GPCRs in vertebrates.

Our approach is conservative and designed to search for novel odors that share structural features from a previously tested odor panel. Odor molecules are limited in size as well, and may offer a limited scaffold such that novel isofunctional chemotype identification may not be as prevalent as has been seen in other examples of scaffold-hopping (*Schneider et al., 2006*). However while compounds that share similar values for the optimized descriptors do have structural similarity for selected parts of the molecule, it does not mean that they are not structurally different in other parts of the molecule. In the future, application of machine learning approaches, such as Support Vector Machines (SVMs) to the receptor-optimized molecular descriptor sets, may be useful to further increase the predictive ability. Additionally, we could replace our SFS approach with sequential floating search techniques, which allows for removal, as well as addition, of descriptors in the growing optimized list.

Our predictions suggest that a number of odorants at low concentrations may be detected by only a few receptors. In the current model of combinatorial coding emphasis is placed on the notion that combinations of several odorant receptors detect the majority of volatile chemicals, with the exception of pheromones and CO2. One possible explanation for this disparity could be that our predictions are fundamentally conservative in nature because we focus only on structurally similar ligands and 7-transmembrane heteromeric receptors may also contain additional unexplored binding sites. Another possibility is that previously tested subsets of odors were potentially selected on the basis of strong responses in electroantennograms and behavior assays, which could bias selection of cross-activating odors. In fact, complex fruit odor blends activate fewer Ors than the number activated by individual odorants at comparable concentrations using electrophysiology (*Hallem and Carlson, 2006*) and Calcium imaging (*Semmelhack and Wang, 2009*). The architecture of the olfactory code therefore appears to integrate two different models. On the one hand, most odors are detected by a few Ors from the repertoire, which may enhance the specificity of the olfactory system for detection of a large number of odors. On the other hand, 15–20% of odors are predicted to activate several Ors (up to 50%) at the same time, which may serve to aid the olfactory of the system in discriminating between fine concentration changes of important stimuli by having Ors tuned to low and high concentrations such as shown for Or42a and Or42b (*Kreher et al., 2008*).

By identifying a large number of new ligands for each odorant receptor, we can also begin to systematically compare the ligand tuning profiles for each in the endogenous neurons vs the 'empty neuron' decoder system. If clear differences were identified, it could enable the identification of underlying reasons such as differences in levels of receptor expression in the neurons, or presence of different odorant binding proteins (OBPs) in the sensillum lymph.

This cheminformatics pipeline can also be applied for system-level analysis of other insects whose receptors and ORNs have been decoded such as mosquitoes (*Carey et al., 2010*), and vertebrates such as mice and humans (*Saito et al., 2009*). The search for novel insect repellents and attractants for species that transmit disease and destroy crops can be greatly assisted by a rational prioritization using such a cheminformatics approach.

## Materials and methods

### Virtual odor compound library

We assembled a subset of 3197 volatile compounds from annotated origins including plants (*Knudsen et al., 2006*), insects (*El-Sayed, 2009*), humans, and a fragrance collection (*Sigma-Aldrich, 2007*) that may have additional fruit and floral volatiles (*Zeng et al., 1991*; *Cork and Park, 1996*; *Zeng et al., 1996*; *Meijerink et al., 2000*; *Curran et al., 2005*; *Knudsen et al., 2006*; *Gallagher et al., 2008*;

*Logan et al., 2008*). We also assembled a subset of 241,150 odors from Pubchem, which have similar characteristics to known odor molecules. Compounds met a criteria of MW <200 and only being composed of the following atoms (C, O, N, H, I, Cl, S, F).

## Calculation of 3D conformations

The three-dimensional structures were predicted for compounds through use of the Omega2 software package (*Bostrom et al., 2003*; *Hawkins et al., 2010*). The Omega2 software package identified the lowest energy 3D conformer for each compound in our Pubchem and Natural compound libraries were stored for use in molecular descriptor calculation.

## Calculation of molecular descriptors

Commercially available software packages Cerius2, Accelrys (200 idescriptors) and Dragon, Talete (3224 descriptors) were used to calculate molecular descriptors from three-dimensional molecular structures. Descriptor values were normalized across compounds to standard scores by subtracting the mean value for each descriptor type and dividing by the standard deviation. Molecular descriptors that did not show variation in values across the compounds were removed. Maximum Common Substructures were determined using an existing algorithm (*Cao et al., 2008b*). Atom Pairs were computed from the version implemented in ChemmineR (*Cao et al., 2008a*).

## Classification of active compounds

Since we were interested in identifying descriptors which best described activating compounds, we needed to first determine which compounds to classify as 'active' based on their electrophysiology activity for the receptor being studied. All of the training odors were clustered using hierarchical clustering by activity individually for each Or. The resulting tree can then be then be used to select the branch containing the majority of activating odors (>50 spikes/s). The activity threshold therefore was set as the lowest spike/s activity of any odor present in the selected branch.

## Determination of Or-optimized descriptor subsets

A compound-by-compound activity distance matrix was calculated using training odor activity data for each of the Ors (*Hallem and Carlson, 2006*). A separate compound-by-compound descriptor distance matrix was calculated using the 3424 descriptor values for training odors calculated by Dragon and Cerius2. Activating compounds for each Or were identified individually through activity thresholds, as described above. The correlation between the compound-by-compound activity (CbCA) and compound-by-compound descriptor distance matrices were compared for each actively classified compound, considering their distances to all other compounds. The goal was to identify molecular descriptors that best correlated with activity. To achieve this we applied a sequential forward selection (SFS) approach to identify optimal descriptors for each Or (*Whitney, 1971*). The SFS functioned by iteratively building a list of molecular descriptors for a single Or by maximally increasing the correlation between the CbCA and CbCD matrices. In the first iteration the values for each single molecular descriptor were used to create CbCD matrices. The rows corresponding to activating compounds were compared to the same rows of the CbCA matrix by correlation. The descriptor which best described the activity (results in the highest correlation between descriptor and activity) was retained. In the second iteration the best single descriptor was combined with all possible descriptors and correlations are calculated again, resulting in a best two-descriptor combination. The process was continued in this fashion to iteratively search for additional descriptors with each iteration aiming to further increases in correlation values. In this manner, the size of the optimized descriptor set increases by one in each iteration, as the best descriptor set from the previous step is combined with all possible descriptors to find the next best descriptor. This process is halted when all possible descriptor additions in an iteration fails to improve the correlation value from the previous step. Molecular descriptors can be selected multiple times for each Or, effectively creating weights for descriptors, as a descriptor that was selected twice will have double the importance when predicting activity of the odor libraries. This whole process is run independently for each Or resulting in unique descriptor sets that are optimized for each Or.

## Calculation of accumulative percentage of actives (APoA)

The accumulative percentage of actives is calculated for each descriptor set individually as previously described (*Chen and Reynolds, 2002*). Compounds are ranked according to their distance from each

known activator using the Or-optimized descriptor values as distances, resulting in one set of ranked compound distances from each activating odor. Moving down the list for each of these rankings, ratios are calculated for the number of activating compounds observed divided by the total number of compounds inspected, or the APoA. APoA values are averaged across all activating compound rankings for each receptor, creating a single set of mean values representing the APoA for a single Or and descriptor set. Using this approach, ApoA mean values are calculated for each of the 24 Ors separately for each descriptor set used, including Or-optimized sets, all Dragon descriptors, all Cerius2 descriptors, Atom Pair, and Maximum Common Substructure. The area-under-the-curve (AUC) scores were calculated by approximation of the integral under each plotted APoA line.

## Clustering Ors by most common descriptors

The first 20 descriptors selected by our optimized descriptor selection algorithm for each Or were used to create an identity matrix. Each row representing an Or and column value specifying the presence of absence of a specific descriptor. This matrix was then converted into an Or-by-Or Euclidean distance matrix and clustered using hierarchical clustering and complete linkage.

## Clustering compounds by activity of Or

The responses of each of the Ors that had previously been tested against a panel of compounds were converted into an Or-by-Or Euclidean distance matrix (*Hallem and Carlson, 2006*). Ors were clustered using hierarchical clustering and complete linkage. Specifically, this was achieved by creating a compound-by-compound distance matrix using the differences in activity between compounds tested on a singe Or. Hierarchical clustering using each Or distance matrix and then identifying the sub cluster which contained the most compounds.

## Clustering Ors by predicted ligand space

Percentages of overlapping predictions within the top 500 predicted compounds were calculated pair-wise for all Ors. Euclidean distances were calculated from the similarity between Ors.

## Calculation of Or prediction distribution frequencies

Initially, all extreme outliers were removed from the dataset for each Or. On average 5.82 compounds were removed for each Or, resulting in a mean dataset reduction of 0.0024%. Next, all compounds whose distance was >3 standard deviations from the strongest activating compound were removed to reduce outliers. Distribution frequencies were produced for each Or. All compound distances were converted into a percentage of the most distant compound for each Or. Frequencies of compounds in the top 15% were plotted.

## Or-ligand interaction map

The Or-ligand interaction map was developed using Cytoscape (*Shannon et al., 2003*). Each predicted Or-ligand interaction from the top 500 predicted ligands for all of the Ors listed were used to calculate the map. All predicted interactions are labeled in purple. In addition all interactions identified in this study and the previous study (*Hallem and Carlson, 2006*) were included and labeled in gray. All compounds are represented as small black circles and Ors are represented as large colored circles. Or names are provided on the upper right corner of each Or.

## Computational validation of *Drosophila* receptor–odor predictions

We performed five independent fivefold cross-validations. For each independent validation the dataset was divided into five equal sized partitions containing roughly 22 compounds each. During each run, one of the partitions is selected for testing, and the remaining four sets are used for training. The training process is repeated five times with each unique odorant set being used as the test set exactly once. For every training iteration, a unique set of descriptors was calculated from the training compound set. These descriptors were then used to calculate distances of the test set compounds to the closest activating compound, exactly as we use to predict ligands in our ligand discovery pipeline. Once test set compounds have been ranked by distance from closest to furthest to a known activating in the training set, a receiver operating characteristics (ROC) analysis is used to analyze the performance of our computational ligand prediction approach. Using ROC we were able to determine our predictive ability for the 12 receptors. This validation could be performed only on receptors for which sufficient

training odors had previously been identified. We consider this to consist of at least one very strongly activating known odor (>150 spikes/s) and at least five strongly activating odors (>100 spikes/s), thus allowing for at least one activating odor for each of the five test sets in the cross-validation (DmOr7a, DmOr9a, DmOr10a, DmOr22a, DmOr35a, DmOr43b, DmOr12, DmOr59b, DmOr67a, DmOr67c, DmOr85b, DmOr98a). Test set validations for all 12 Ors were combined and a single ROC curve representing an average across all Ors was plotted (*Figure 1C*).

## Electrophysiology

Extracellular single-sensillum electrophysiology was performed as before (*Dobritsa et al., 2003*; *Hallem and Carlson, 2006*; *de Bruyne et al., 2001*) with a few modifications. Diagnostic odorants were used to distinguish individual classes of ORNs in sensilla (ab1-ab7) and therefore unequivocally identify the target Or expressing ORN for testing (*de Bruyne et al., 2001*; *Hallem et al., 2004*). 50 µl odor at $10^{-2}$ dilution in paraffin oil was applied to cotton wool plugged odor cartridge. Due to variability in temporal kinetics of response across various odors, the counting window was shortened to 250 ms from the start of odor stimulus.

## Acknowledgements

We would like to thank Thomas Girke and Y Cao for assistance with cheminformatics; Jocelyn Millar for providing chemicals and Anupama Dahanukar for critical reading of the manuscript. SMB is supported by an NSF IGERT grant in Chemical Genomics.

## Additional information

### Competing interests

SMB: Listed as an inventor in patent applications filed by the University of California, Riverside. AR: Holds equity in an insect research company (Olfactor Labs) and is listed as an inventor in patent applications filed by the University of California, Riverside. The other author declares that no competing interests exist.

### Funding

| Funder | Grant reference number | Author |
| --- | --- | --- |
| National Science Foundation | IGERT | Sean Michael Boyle |

The funder had no role in study design, data collection and interpretation, or the decision to submit the work for publication.

### Author contributions

SMB, AR, Conception and design, Acquisition of data, Analysis and interpretation of data, Drafting or revising the article; SM, Acquisition of data, Analysis and interpretation of data

## Additional files

### Supplementary files

• Supplementary file 1. (**A**) Optimized descriptor sets for each *Drosophila* Or. Optimized descriptors occurrences, symbol, brief description, class, and dimensionality are listed. A summary of the total number of descriptors selected for the receptor repertoire is provided at the beginning. Descriptors are listed in ascending order of when they were selected into the optimized set, such that the descriptors selected first are more important. Weights indicate the number of times a descriptor was selected in an optimized descriptor set. (**B**) Top 100 predicted compounds for each *Drosophila* Or. Chemical name or Pubchem compound ID (CIDs), SMILES strings, and distances, of the top ~100 predicted compounds for each Or. All distances represent the minimum distance based on optimized descriptors to the previously known strongest active compound listed in the gray cells for that particular Or.

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
