## [Decision Letter]

Thank you for sending your work entitled “Expanding the olfactory code by in silico decoding of odor-receptor chemical space” for consideration at *eLife*. Your article has been favorably evaluated by a Senior editor and 3 reviewers, one of whom is a member of our Board of Reviewing Editors.

The Reviewing editor and the other reviewers discussed their comments before we reached this decision, and the Reviewing editor has assembled the following comments to help you prepare a revised submission.

The manuscript by Boyle et al. describes the use of a computational approach to identify new receptor-odor pairs for *Drosophila* based on analysis of a previously published small pool of known agonists and antagonists on a panel of 19 Drosophila odorant receptors. The achievements include: 1) by validating 71% of the 141 experimentally tested odor-receptor pairs, the authors expanded the existing dataset by a significant amount (it would be useful for the authors to determine the precise number and add to the revised text); 2) the 19 Ors **×** top 500 odorants matrix could constitute a much larger odor-receptor pairs that have a good chance to be authentic; 3) the same method can in principle be applied to other organisms, after experimentally determining a reasonably rich matrix of receptor-odor pairs. Overall the manuscript is of high technical quality. We would therefore like to invite the authors to revise the manuscript by addressing the following specific critiques.

1) The authors should make more explicit the limitations of the study in the Abstract and conclusions; their current summary creates false expectation. First, they can only predict more ligands based on receptors that have already been experimentally tested against a large number of ligands. Second, they predicted ligands for only 19 Ors; although technically it may constitute the “majority” of ORNs that utilize Ors, it is certainly not a “majority” of ∼50 ORN types in adult Drosophila (as a significant fraction of ORNs utilize Irs). Third, the validated 71% came from only 9 Ors; it is unclear whether this can be generalized to the other untested Or classes. Fourth, there is no evidence that any of the agonists or antagonists identified in silico are better (higher affinity) or more likely to be the true biologically relevant ligands for these receptors than those identified in small chemical libraries based on ecologically reasoning (i.e., the training set).

2) The use of the optimized descriptor sets need to be defined more explicitly. In Table 1 what are the numbers given for each descriptor? If these are classes of descriptors then how many descriptors from the original set of over 3,000 are actually being used, and what are they? And how do they appear to be relevant to odor quality? One of the issues with the similar analysis performed by Haddad was that the descriptors (there determined by a PCA analysis) seemed to have little obvious relevance to odor quality. Is the optimized method presented here an improvement on that? The description of the SFS approach does not provide any detail as to how each incremental descriptor was chosen to grow the set. Was this done by a purely statistical or machine learning method or did the investigators use some intuition regarding the likelihood of the descriptor to be relevant to odor character?

3) Regarding the paragraph ‘Producing a systems level view of receptor activity for the *Drosophila* antenna’, either this very intriguing analysis should go in the Discussion or a deeper analysis is required for the reader. The network drawn in Figure 6A is impossible to interpret. The authors might choose a natural blend, or even an artificial one, and show us the network for that particular group of odors. That would be clearer and more useful, especially if it shows something new or unexpected. Showing all the interactions creates an attractive graphic, but not one that is informative.

4) The authors should provide more raw data from their analyses.

---

## [Author Response]

*1A) “…they can only predict more ligands based on receptors that have already been experimentally tested against a large number of ligands.*”

We agree with the critique that we are only able to predict ligands for which a training odor set has been made available. We had included this information within the Results section, but have now incorporated this into our Abstract and Discussion sections.

*1B) “…they predicted ligands for only 19 Ors; although technically it may constitute the “majority” of ORNs that utilize Ors, it is certainly not a “majority” of ∼50 ORN types in adult Drosophila (as a significant fraction of ORNs utilize Irs).*”

We agree that the wording could come across as ambiguous, which is not our intention. We have modified the text to clarify the distinction and included a short discussion of potential for predictions from other classes of chemoreceptors in the Discussion section.

*1C) “…the validated 71% came from only 9 Ors; it is unclear whether this can be generalized to the other untested Or classes.*”

The rationale for selecting the 9 receptors for testing was accessibility to electrophysiology and unambiguous identification using a diagnostic odor panel. We have modified the text in the Abstract to accurately present the experimental data and clarify that 9 Ors were validated.

*1D) “…there is no evidence that any of the agonists or antagonists identified in silico are better (higher affinity) or more likely to be the true biologically relevant ligands for these receptors than those identified in small chemical libraries based on ecologically reasoning (i.e., the training set).*”

We agree with the critique. Our focus in this analysis was to create a method that could identify a large number of active compounds, which we were successful at. We expect this will be useful for hypothesis generation and potentially identifying stronger, or ecologically and behaviorally important odors. We anticipate that availability of large number of candidate ligands will also be useful in behavioral disruption programs for pest and disease vector species, since it will allow a researcher to judiciously select affordable, pleasant smelling and environmentally safe chemicals for applications.

*2A) “In*
Table 1
*what are the numbers given for each descriptor? If these are classes of descriptors then how many descriptors from the original set of over 3,000 are actually being used, and what are they? And how do they appear to be relevant to odor quality?*”

The numbers in Table 1 represent the total number of molecular descriptors from these classes that were identified by our approach as an overview. We agree that it would be much more informative to provide the exact molecular descriptor sets that were optimized for each receptor. We have now created a new table that provides molecular descriptor symbols, weights, descriptions, classes, and descriptive dimensionality for each Or-optimized set used in our study. This provides a wealth of useful data. While several of these descriptors are very specific and represent high dimensional graph based theory, a number of selected descriptors are easily understood such as functional group counts and atom types descriptors. Through this table readers will be able to identify which functional groups and 2D fingerprints are most important for a particular receptor and specialists will be able to utilize them in their own analysis such as the prediction pipeline we have created.

*2B) “The description of the SFS approach does not provide any detail as to how each incremental descriptor was chosen to grow the set.*”

We apologize for being unclear and have clarified this in the Materials and methods section.

*3) “…this very intriguing analysis should go in the Discussion or a deeper analysis is required for the reader.*”

We agree and thank you for your suggestion. We have moved this section to the Discussion.

*4) “The authors should provide more raw data from their analyses.*”

We thank you for this suggestion and agree that this manuscript would benefit from increased raw data. We have incorporated several new supplemental tables and figures. Newly incorporated data includes: optimized molecular descriptor sets for each predicted Or, including name, weight, class, description, and dimensionality; top 100 predicted compounds for each of the receptors analyzed; APoA plots for individual Ors; and pharmacophore structures for active compounds for each Or.